# Antithrombotic Therapy Following Transcatheter Aortic Valve Replacement

**DOI:** 10.3390/jcm11082190

**Published:** 2022-04-14

**Authors:** Camille Granger, Paul Guedeney, Jean-Philippe Collet

**Affiliations:** ACTION Study Group, UMR_S 1166, Institut de Cardiologie, Sorbonne Université, Pitié Salpêtrière Hospital (APHP), 75013 Paris, France; cmll.granger@gmail.com (C.G.); paul.guedeney@aphp.fr (P.G.)

**Keywords:** TAVR, antithrombotic therapy, oral anticoagulation, DOAC, VKA, SAPT

## Abstract

Due to a large technical improvement in the past decade, transcatheter aortic valve replacement (TAVR) has expanded to lower-surgical-risk patients with symptomatic and severe aortic stenosis. While mortality rates related to TAVR are decreasing, the prognosis of patients is still impacted by ischemic and bleeding complications, and defining the optimal antithrombotic regimen remains a priority. Recent randomized control trials reported lower bleeding rates with an equivalent risk in ischemic outcomes with single antiplatelet therapy (SAPT) when compared to dual antiplatelet therapy (DAPT) in patients without an underlying indication for anticoagulation. In patients requiring lifelong oral anticoagulation (OAC), the association of OAC plus antiplatelet therapy leads to a higher risk of bleeding events with no advantages on mortality or ischemic outcomes. Considering these data, guidelines have recently been updated and now recommend SAPT and OAC alone for TAVR patients without and with a long-term indication for anticoagulation. Whether a direct oral anticoagulant or vitamin K antagonist provides better outcomes in patients in need of anticoagulation remains uncertain, as recent trials showed a similar impact on ischemic and bleeding outcomes with apixaban but higher gastrointestinal bleeding with edoxaban. This review aims to summarize the most recently published data in the field, as well as describe unresolved issues.

## 1. Introduction

Since the first implantation in an inoperable patient, by Alain Cribier in 2002 [1], transcatheter aortic valve replacement (TAVR) has become a first-line treatment option for patients with symptomatic and severe aortic stenosis. With the continuing improvement of devices and operators’ increased experience, TAVR is now recommended for patients at either high or intermediate surgical risk who are suitable for trans-femoral approach [2,3], and since 2019 its use exceeds surgical aortic valve replacement (SAVR) [4,5]. First approved by the Food and Drug Administration (FDA) in high-surgical-risk patients in 2011, it was extended to intermediate-risk patients in 2016 and low-surgical-risk patients in 2019, following the results of two large randomized controlled trials (RCT) which showed TAVR efficacy and safety compared to the surgical approach [6,7]. Nevertheless, the long-term prognosis after TAVR-procedure is mainly impacted by ischemic and hemorrhagic complications. Therefore, antithrombotic therapy after TAVR is essential to prevent embolic complications such as stroke, but it needs to be well evaluated considering the risk of major and life-threatening bleeding, especially in frail patients at high-surgical-risk who often have comorbidities increasing the risk of such adverse events. For the first time in recent decades, international guidelines on the matter of antithrombotic therapy following TAVR are no longer based on expert consensus but are evidence-based, following the publication of recent dedicated RCTs. This review aims to describe the most recent findings in the field, as well as unresolved issues.

## 2. Thrombo-Embolic Risk after TAVR

With the improvement of procedural technique and extension to lower-surgical-risk patients, mortality rates associated with TAVR kept decreasing in the last decade. Nonetheless, available data on long-term outcomes after TAVR from the first trials report a high rate of long-term mortality (28 to 72% at 5 years) (Figure 1). Mortality and morbidity after TAVR are mainly related to bleeding and ischemic complications, particularly stroke.

### 2.1. Stroke

Stroke is one of the most dreaded complications after TAVR and is associated with an increased rate of mortality [8,9,10]. The incidence of major or disabling stroke ranges from 0 to 5% at 30 days and 0.2 to 7.8% at 1 year in the pivotal trials [6,7,11,12,13,14,15,16] (Figure 2). While recent RCTs of low-surgical-risk patients showed an important decrease of 1-year stroke rates when compared with high-risk patients (0.2 to 2.9% [6,7] vs. 5.1% to 7.8%, respectively [11,12,13]), only a small and slow downward trend was observed in stroke rates in the STS-ACC TVT (Society of Thoracic Surgeons-American College of Cardiology Transcatheter Valve Therapy) Registry from 2011 to 2019, when considering all-risk patients [4]. The incidence of 30-day and 1-year stroke is approximately 2.0–2.5% and 4%, respectively [4,10,17]. In a large real-world retrospective cohort, the incidence of transient ischemic attack (TIA) and stroke at 30 days was 0.4% and 2.3%, respectively, with stroke associated with a higher rate of 30-day mortality [10]. Indeed, perioperative strokes after TAVR are associated with a 6-fold increase in 30-day mortality, with an incidence between 12% and 20% [17,18,19]. If 30-day and 1-year stroke risk is now well-established in TAVR patients, few data are available on cerebrovascular events beyond the first year after the intervention. In a large cohort recently published, stroke rate remained stable between 2–3% per year after eight years of follow-up [19]. In randomized trials, major or disabling stroke has been reported with an incidence between 2.6% and 6.8% at two years [16,20,21], and up to 10–12% at five years [20,22]. Consistently, in the NOTION trial, 8.3% of low-surgical-risk patients experienced stroke after long-term follow-up of 8 years [23]. In a recent prospective study, 5.1% of patients presented late cerebrovascular events (LCVEs) (>30 days after TAVR) within 2 years after TAVR, among which 70% had disabling strokes, with the occurrence of stroke associated with an in-hospital mortality rate of 29.2% [24]. While the deleterious impact of clinical stroke is grounded, the prognosis of silent brain injury (SBI) remains unclear. Indeed, many reports have now demonstrated the risk of silent cerebrovascular events. Although the incidence of clinical stroke is relatively low, SBI is observed in more than 70% of patients undergoing cerebral magnetic resonance imaging (MRI) after TAVR (71–84%) [25,26,27,28]. In non-cardiac studies, the presence of SBI on MRI has been associated with an increased risk of dementia and cognitive decline [29]. Current knowledge is scarce on the consequences of SBI after TAVR. If most of SBI disappear within a few months on repeated MRI after TAVR, their occurrence has been associated with a limited but significant deterioration of early neurocognitive function [25,27]. Robust evidence with long-term follow-up studies remains necessary to assess the clinical impact of SBI in TAVR patients. Identifying predictors of cerebral embolization during and after TAVR will provide a better comprehension of prevention strategies to reduce its incidence. More than 90% of cerebrovascular events after TAVR have an ischemic origin, with most of them from an embolic source [10]. Histopathological studies have shown that emboli during TAVR consisted of thrombus, calcification and tissue from the native aortic valve leaflets or aortic wall [30,31,32]. Balloon post-dilatation of the valve prosthesis or valve embolization are predictors of acute CVEs [33]. Therefore, protecting the brain from emboli during TAVR remains a challenge. The risk of thromboembolic events is highest within 24–48 h after the procedure [9,34], but seems more related to technical aspects of the procedure than to the periprocedural antithrombotic therapy. Indeed, in the BRAVO-3 (Effect of BivaliRudin Aortic Valve Intervention Outcomes 3) trial, use of bivalirudin versus heparin anticoagulation during TAVR was not associated with differences in procedural cerebral embolization nor 30-day mortality [35]. Uncertainty remains on which best antithrombotic regimen between antiplatelet therapy and OAC should be given after TAVR to prevent cerebral embolic events, with further studies warranted.

### 2.2. Atrial Fibrillation and the Risk of Cerebrovascular Events

Atrial fibrillation (AF) is common in patients undergoing TAVR and is associated with worse outcomes. Both pre-existing AF and new-onset AF are predictors of adverse ischemic cerebrovascular events, mortality and bleeding post-TAVR [33,36,37,38,39]. Pre-existing AF is present in 30–40% of patients undergoing TAVR [6,7,11,12,13,14,15,16,37,38] (Figure 3). While new onset of AF is less frequent in patients with severe aortic stenosis undergoing TAVR compared to SAVR [40,41], it is a common complication after TAVR, with a prevalence reaching 16% at 30 days, 21% at 1 year and 25% at 5 years (Figure 3) [14,42], and is associated with a higher rate of stroke and mortality within the first year [37,43,44,45,46,47]. Despite the known risk of ischemic and embolic events with AF, there is still an important international variability in discharged medication patterns in patients with a history of AF undergoing TAVR [44], with 40% of patients discharged without OAC therapy in the recent large registry from Sherwood et al. [48]. Thus, clarification is needed on the antithrombotic regimen in patients who underwent TAVR with prior or new onset of AF, and will further be discussed.

### 2.3. What Is the Current Place of Cerebral Embolic Protecting Devices?

Cerebral embolic protecting devices (EPD) are now approved by the FDA, but only a few randomized studies have evaluated their ability to reduce stroke and mortality which remains controversial. Small randomized controlled trials demonstrated a reduction of the number of new cerebral lesions and their volume on MRI with cerebral EPD during TAVR, but they were underpowered to demonstrate a significant decrease in stroke rates and mortality [49]. While use of the Sentinel EPD during TAVR in the SENTINEL trial led to capturing embolic debris in up to 99% of patients, it had no impact on neurocognitive function [31]. In contrast, the MISTRAL C trial showed fewer lesions and smaller lesion volume in patients with the Sentinel EPD, and a decrease in neurocognitive deterioration as compared to unprotected procedures (4% vs. 27%, respectively; *p* = 0.017) [50]. Consistently, the number and volume of new brain lesions were significantly reduced during TAVR with cerebral EPD in the CLEAN TAVI trial [51]. Only observational studies suggested a significant reduction of early (<30 days after TAVR) stroke and mortality rates in patients undergoing TAVR with cerebral EPD, compared to unprotected procedures [52,53,54]. A recent large real-world study showed significantly lower mortality (0.5% vs. 1.3%, *p* < 0.01) and lower ischemic stroke (1.4% vs. 2.2%, *p* < 0.01) with cerebral EPD vs. no cerebral EPD, respectively [55]. Similar results were reported in a meta-analysis including three RCTs and four observational studies, in which patients undergoing TAVR with the Sentinel EPD experienced a lower rate of stroke, 30-day mortality and bleeding compared with patients without EPD [56]. In contrast with these results, two meta-analysis showed a reduction of early stroke with cerebral EPD during TAVR but without impact on in-hospital mortality [57,58]. Thus, the use of EPD during TAVR seems safe, providing a reduction on new lesion volume on MRI, but its efficacy on clinical outcomes remains controversial and is only suggested in observational studies. Two large ongoing randomized trials will help clarify this issue, evaluating the efficacy of the Sentinel cerebral EPD on the rate of stroke within 72 h after TAVR: PROTECTED TAVR [Stroke PROTECTion With SEntinel During Transcatheter Aortic Valve Replacement], (NCT04149535, *N* = 3000) with an estimated completion date of July 2022, and BHF PROTECT-TAVI (British Heart Foundation Randomised Trial of Routine Cerebral Embolic Protection in Transcatheter Aortic Valve Implantation) (ISRCTN16665769, *N* = 7730), with an estimated completion date of April 2026.

### 2.4. Myocardial Infarction, Periprocedural Myocardial Injury and Concomitant Coronary Artery Disease

Myocardial infarction (MI) is a rare complication of TAVR, with reported rates from 0 to 2.8% at 30 days and 0.4 to 3.5% at 1 year (Figure 4) [6,7,11,12,13,14,15,16], according to the definition of the Valve Academic Research Consortium 2 (VARC-2) consensus document. This prior VARC-2 consensus document defined MI in patients undergoing TAVR as new ischemic symptoms or signs occurring within 72 h after TAVR associated with an elevation in cardiac biomarkers (peak value exceeding ≥ 15-fold upper limit normal (ULN) for troponin or ≥5-fold ULN for creatine kinase MB (CK-MB)) [59]. Despite the relative low rate of VARC-2 defined MI after TAVR, a large majority of patients have a significant isolated elevation in cardiac biomarkers within 72 h after the procedure, matching post-procedural myocardial injury (PPMI) definition. Indeed, PPMI ranges from 20 to 79% in observational reports [60,61,62,63], but its impact on clinical outcome remains controversial. Some cohorts showed no association of PPMI with 30-day and long-term mortality [62,63,64], while others reported a significant relationship between PPMI and mortality rates [60,61,64]. Likewise, two meta-analyses gathering a large number of patients showed a significant association of PPMI after TAVR with both 30-day and long-term all-cause mortality [65,66]. It has also been suggested that a cardiac biomarker elevation lower than the VARC-2 defined limit would be associated with an increase in mortality after TAVR [60,67]. The trans-apical approach, major procedural complications, old valve generations and total rapid pacing time have been associated with an increased risk of PPMI [60,68,69,70,71]. One recent study found anticoagulation treatment to be significantly associated with a reduction of PPMI [69], but insufficient data are available on the association of OAC with the risk of myocardial injury and myocardial infarction during and after TAVR. Furthermore, the risk of MI increases in patients with concomitant coronary artery disease (CAD), which is highly frequent in patients undergoing TAVR (50–60%) [69,72,73,74]. Compared to a surgical approach with SAVR and coronary artery bypass grafting (CABG), a complete percutaneous approach with TAVR and PCI seems safe with equivalent results on long-term mortality and cardiovascular outcomes in patients with severe aortic stenosis and concomitant CAD [75,76]. The data remains controversial on the clinical impact and prognosis significance of CAD in TAVR patients [61,73,77,78,79,80,81,82], and the best revascularization strategy to adopt is still unclear. The European Society of Cardiology (ESC) guidelines recommend PCI for coronary artery diameter stenosis > 70% in proximal segments (Class of recommendation: IIa; Level of evidence: C) in patients undergoing TAVR with concomitant CAD, with dual antiplatelet therapy (DAPT) recommended after PCI with a duration established according to bleeding risk [2] (Table 1). Prior PCI seems to have a beneficial impact on the long-term prognosis of TAVR patients with concomitant CAD [69], as incomplete revascularization has been associated with a higher rate of mortality [79,82,83,84]. There is a lack of robust data on the optimal strategy and timing of PCI among TAVR patients with concomitant CAD, since only one underpowered randomized trial evaluated the impact of prior PCI in this population [85].

The definition and classification of MI has recently been updated in a consensus document and is now defined consistently as proposed by the Fourth Universal Myocardial Definition [86]. Thus, spontaneous MI (>48 h after the procedure) is defined by new ischemic signs or symptoms associated with a rise and/or fall of cardiac troponin (cTn) values with at least one value above the 99th percentile upper reference limit (URL). For periprocedural MI (within 48 h after the procedure), VARC-3 proposes to use the modified SCAI (Society for Cardiovascular Angiography and Interventions) and ARC-2 (Academic Research Consortium-2) classification, defined by a rise of creatinine kinase MB (CK-MB) ≥10-fold upper limit normal (ULN) or rise of cTn to ≥70-fold ULN; or an association of at least one ischemic sign and a peak value exceeding ≥ 5-fold ULN for CK- or ≥35-fold ULN for cTn [87].

### 2.5. Valve Thrombosis

While symptomatic valve thrombosis is scarce after TAVR, with an incidence between 0.6 and 2.8% [91,92,93], many studies now reported that a large number of patients had subclinical leaflet thrombosis (SLT) with transcatheter and surgical bioprosthetic aortic valves, and SLT prevalence is higher with transcatheter than with surgical valves [94,95,96,97]. SLT is characterized by a thin layer of thrombus on the leaflet of the bioprosthesis, revealed on high resolution 4-dimensionnal (4D) cardiac computed tomography (CT) by hypo-attenuated leaflet thickening (HALT). HALT can be associated with either normal or reduced leaflet motion (RLM), and is classified as hypo-attenuation affecting motion (HAM) when more that 50% of leaflet motion is affected. After TAVR, SLT frequency ranges from 15 to 38% [92,98,99,100,101,102]. Three recent RCTs showed a prevalence of HALT of 5–17% at 30 days, 23% at 3 months and 20–30% at 1 year, with RLM found in up to 31% at 1 year [95,99,103]. As opposed to symptomatic valve thrombosis revealed by dyspnea, heart failure or embolic events, SLT is often an incidental finding with controversial clinical impact. The presence of SLT has been associated with the occurrence of cerebrovascular events [94,95,97,104,105]. In a recent meta-analysis, patients with SLT diagnosis at follow-up had a 2.6-fold increased risk of stroke and TIA when compared to patients without SLT [98]. No association has been established between SLT and mortality rates in meta-analysis and large observational studies [92,94,105]. Conversely, in the Evolut Low Risk sub-study, in which one-third of patients had HALT and RLM at 1 year, no evidence of a relationship between these leaflet abnormalities and adverse ischemic clinical events was highlighted [103]. HALT is a dynamic process, with spontaneous regression in up to 50% of patients at 1 year, and new development in 20% of patients at 1 year among patients with no prior HALT on early cardiac CT scan [95,103]. Its physiopathology is not yet fully understood and the management of the antithrombotic regimen in the prevention of SLT remains uncertain. As thrombus formation starts within a few hours after the intervention [106], antithrombotic therapy during and after TAVR aims to prevent valve thrombosis and ischemic-related outcomes. In a histological study, thrombus has been found in all explanted valves, with evolution to fibrosis after 60 days and calcification within 4 years, which contributed to progressive leaflet thickening and valve deterioration [106]. Indeed, it has been suggested that SLT might accelerate the progression towards valvular hemodynamic deterioration (VHD), with an increased mean trans-aortic gradient [97,102,107,108]. However, these findings have not been reported in recent small sub-studies of RCTs [95,103], and further research is needed to assess the association of SLT and risk of VHD. Identifying its predictors is a key point for the assessment of prevention strategy of valve thrombosis. Patients with smaller aortic annular areas often receive comparable size valves as commonly sized annular areas, with valve oversizing by more than 20% associated with an increased risk of HALT (OR 23.5, *p* = 0.006) [109]. Valve-in-valve procedures, balloon-expandable TAVR, large sinus of Valsalva and severe patient-prosthesis mismatch are risk factors of valve thrombosis [93,110,111]. While clinical valve thrombosis requires anticoagulant treatment [2,112], uncertainty remains on the clinical implication and the need for care in SLT. Antiplatelet therapy does not seem efficient enough to prevent SLT, as OAC has been associated with a lower incidence of SLT than with single or dual antiplatelet therapy [98,104], with a relative risk reduction of up to 58% with anticoagulation therapy when compared to antiplatelet therapy [98]. In addition, switching from antiplatelet to anticoagulation therapy seems effective in SLT resolution [96,98]. In the study by Chakravarty et al., both NOAC and warfarin, but not antiplatelet therapy, were equally effective in the prevention and treatment of SLT [97]. Furthermore, in some observational studies, a lack of OAC exposure post-TAVR seems to correlate with a greater risk of VHD [107,113,114]. The recent GALILEO (Global Study Comparing a Rivaroxaban-based Antithrombotic Strategy to an Antiplatelet-based Strategy After Transcatheter Aortic Valve Replacement to Optimize Clinical Outcomes) 4D sub-study demonstrated in TAVR patients without an indication for long-term anticoagulation that a treatment with low-dose rivaroxaban plus aspirin was associated with a significant reduction of subclinical leaflet-motion abnormalities when compared to DAPT on 3-month CT (2.1% vs. 10.9%, respectively; *p* = 0.01). However, these findings did not correlate with clinical improvement as mortality, thromboembolic complications and bleeding risks were higher with rivaroxaban in the main trial compared to antiplatelet therapy [99]. Consistently, in the 4D-CT sub-study of the ATLANTIS trial, apixaban was associated with lower valve leaflet thrombosis at 3 months when compared to antiplatelet therapy (8.9% vs. 15.9%, respectively, *p* = 0.011), but without a significant improvement of clinical outcomes [115]. In the recently published ADAPT-TAVR (Anticoagulant Versus Dual Antiplatelet Therapy for Preventing Leaflet Thrombosis and Cerebral Embolization After Transcatheter Aortic Valve Replacement) trial, 229 patients without an indication for OAC who underwent successful TAVR were randomized to compare the effects of 6 months edoxaban to 6 months DAPT (aspirin plus clopidogrel) on leaflet thrombosis assessed by 4D-CT scan. No significant difference was reported in the incidence of leaflet thrombosis on 4D-CT scan between patients treated with edoxaban and DAPT (9.8% vs. 18.4%, respectively; absolute difference 8.5%, RR 0.53, *p* = 0,076). The incidence of new cerebral thromboembolism lesion on brain MRI, and new development of neurological dysfunction were similar between the two groups [116]. Therefore, identification and therapeutic care of SLT remains an issue. Although its use might be effective in the prevention and treatment of SLT, anticoagulation therapy has been associated with an increased risk of mortality and major bleeding events in TAVR patients with no underlying indication for OAC in recent RCTs. Thus, the routine use of OAC should not be recommended to prevent valve thrombosis in high-risk patients without another indication for anticoagulation. Further research is needed to help understanding the clinical impact and optimum prevention strategy of silent leaflet thrombosis.

## 3. Bleeding Events after TAVR

While bleeding risk is less frequent after transcatheter compared to surgical aortic valve replacement [117], it remains a major complication during and after TAVR. Bleeding events are categorized as early (<30 days post-TAVR) and late (>30 days post-TAVR) complications. The prior VARC-2 consensus document divided bleeding events into three types, following the Bleeding Academic Research Consortium (BARC) classification: minor bleeding (BARC 2 and 3a depending on severity), major bleeding (BARC 3a) and life-threatening or disabling bleeding (BARC 3b, 3c, 5) [59]. Recently, the classification of bleeding events after TAVR has been updated in the VARC-3 consensus document, and is now divided into four types: type 1 (minor: BARC 2 and 3a depending on severity), type 2 (major: BARC 3a), type 3 (life-threatening: BARC 3b, 3c and 4) and type 4 (leading to death: BARC 5) [87]. Most of the time, bleeding complications appear within the first 30 days post-procedure [118]. In the pivot trials, early life-threatening and disabling bleeding rates (VARC-2 defined) ranged from 9 to 17% in high-surgical-risk patients, but decreased to 10–12% in intermediate-risk patients and 2–11% in low-risk patients. Bleeding risk remains long after TAVR, with a 1-year severe bleeding incidence from 2.4 to 22.3% (Figure 5) [6,7,11,12,13,14,15,16]. Not only related to fewer comorbidities in lower-surgical-risk patients, the decrease in bleeding rates noticed over the years is associated with an increase in the operator’s experience, reduction of sheath catheter diameter and the use of trans-femoral access rather than trans-apical access where possible. Data from the PARTNER trial reported major late bleeding events in 6% of patients, mainly from gastrointestinal (40%) and neurological (15%) origin [119]. Both early and late bleeding events are associated with poor outcomes and higher 30-day and 1-year post-TAVR mortality rates [120,121,122]. Indeed, in a recent large register comprising more than 200,000 patients, major bleeding was associated with significant higher in-hospital mortality rates (14.4% vs. 4.2%, major bleeding group vs. no major bleeding group, respectively; *p* < 0.01) [123]. Consistently, major or life-threatening bleeding has been associated with a 4-fold increase of 30-day and 1-year mortality after TAVR [119,120]. There is no doubt that bleeding is associated with poor outcomes and that a better assessment of the risk profile is a key step in the refinement of a subsequent antithrombotic regimen. Periprocedural bleedings are mainly access-sites related and associated with pre-existing peripheral arterial disease, sheath diameter and the failure of the percutaneous closure device [124,125,126]. Indeed, trans-apical access has been associated with an 83% increased risk of bleeding [120], but is insignificant nowadays and is used in less than 0.5% patients [10]. Chronic kidney disease, acquired reversible von Willebrand factor deficiency and acquired thrombocytopenia have been identified as risk factors of early bleedings, while age, comorbidities and chronic antithrombotic regimen keep a high bleeding risk late after the procedure [125]. Therefore, the choice of antithrombotic therapy and timing of its administration remains an essential matter for patients undergoing TAVR.

## 4. Periprocedural Antithrombotic Therapy

Bleeding events have a dreadful impact on TAVR patients and mostly occur within the 30 days post-intervention with great risk during the procedure, as well as ischemic stroke and TIA. Optimal periprocedural antithrombotic strategy is the first key step. The BRAVO-3 trial showed no reduction of major thromboembolic events (4.1% vs. 4.1%, *p* = 0.97), but a higher rate of major vascular complications (11.9% vs. 7.1%, *p* = 0.02) with pre-loading clopidogrel on top of aspirin prior to TAVR [127]. Low-dose aspirin alone is the treatment of choice, usually started pre-TAVR, in patients with no indication of OAC [2]. 

Parenteral anticoagulation therapy with unfractionated heparin (UFH) is routinely given to prevent periprocedural thromboembolism, particularly stroke. In the BRAVO-3 trial, bivalirudin did not reduce major bleeding or adverse cardiovascular events at 48 h when compared with UFH [128]. In 2012, an expert consensus document recommended heparin administration to maintain an activated clotting time (ACT) >300 s [89]. More recently, the consensus document of the ESC and the European Association of Percutaneous Cardiovascular Intervention (EAPCI) recommend the use of UFH with an ACT between 250 and 300 s to prevent catheter thrombosis and thromboembolism, with bivalirudin an option if there is prior evidence of heparin-induced thrombocytopenia [129]. Baseline ACT-guided heparin administration has been shown to reduce major bleeding during transfemoral TAVR [130]. Expert consensus documents also suggest that protamine sulfate can be used to reverse anticoagulation before closure to reduce vascular access-site complications and bleeding. Indeed, a significant decrease of major and life-threatening bleeding is obtained with protamine sulfate after TAVR, without a rise in the occurrence of stroke and MI [131].

Few observational studies have evaluated the continuation of OAC during TAVR in patients requiring long-term anticoagulation. Compared to pre-TAVR interruption of OAC, continuation of OAC throughout the intervention did not increase in-hospital or 30-day bleeding rates, nor vascular complications [132,133,134]. An observational cohort reported 30-day major or life-threatening bleeding was not significantly higher in patients with continuation versus interruption of anticoagulation (11.3% vs. 14.3%, respectively, OR 0.86, *p* = 0.39), as well as major vascular complications (11.0% vs. 12.3%, OR 0.89, *p* = 0.52) [132]. Randomized controlled trials, such as the POPular PAUSE TAVI (Periprocedural Continuation Versus Interruption of Oral Anticoagulant Drugs During Transcatheter Aortic Valve Implantation) trial (NCT04437303), are awaited in order to confirm these findings. 

## 5. Antithrombotic Therapy after TAVR

### 5.1. Antiplatelet Therapy after TAVR: Updated Guidelines in 2021

Since 2020, the American College of Cardiology/American Heart Association (ACC/AHA) guidelines recommend a single antiplatelet therapy of aspirin (75–100mg daily) after TAVR in the absence of other indications for oral anticoagulants (class of recommendation: 2a, level of evidence: B-R), while dual antiplatelet therapy (aspirin 75–100 mg plus clopidogrel 75 mg daily) for 3 to 6 months has been retroceded to class of recommendation 2b [3] (Table 1). Indeed, after years of debate, recent RCTs showed no advantage of DAPT when compared to SAPT in patients undergoing TAVR with no indication of OAC and no prior coronary stenting. Two small-scale RCTs did not report difference between SAPT and DAPT after TAVR on ischemic outcomes in this population [135,136]. Consistently, the ARTE (Aspirin Versus Aspirin plus Clopidogrel as Antithrombotic Treatment Following TAVI) trial reported no difference between SAPT and DAPT in the occurrence of death, stroke or TIA at 3 months post TAVR, whereas DAPT was associated with a higher rate of major or life-threatening bleeding events (10.8% vs. 3.6% in the SAPT group, *p* = 0.038) [137]. These results were corroborated by the recent Cohort A of the POPular TAVI (Antiplatelet Therapy for Patients Undergoing Transcatheter Aortic Valve Implantation) trial, in which 665 patients with no indication for OAC were randomized to receive either aspirin alone or aspirin plus clopidogrel for 3 months after TAVR. Bleeding events and the composite end point of bleeding or thromboembolic events at one year were significantly less frequent with aspirin alone than with DAPT (15.1% vs 26.6%, respectively, relative risk (RR) 0.57; 95% CI: 0.42-0.77; *p* = 0.001 for bleeding; 23.0% vs 31.1% *p* <0.001 for noninferiority; RR = 0.74 *p* = 0.04 for superiority, for composite of bleeding or thromboembolic events) [138]. In addition, 2 recent meta-analyses reported lower bleeding events with aspirin alone when compared to DAPT after TAVR without significant differences in mortality, myocardial infarction or stroke [139,140]. Thus, these studies demonstrate no difference between SAPT and DAPT in preventing thromboembolic outcomes after TAVR in patients with no indication of OAC, but a consistent and significant increase in major bleeding events with DAPT. Taking these results into consideration, guidelines from the ESC and the European Association for Cardio-Thoracic Surgery (EACTS) on valvular heart disease were updated in 2021, and from this point, a lifelong single antiplatelet therapy with aspirin (75–100 mg daily) or clopidogrel (75 mg daily) is recommended after TAVR in patients with no baseline indications for OAC (class of recommendation: I, level of evidence: A). DAPT with low dose aspirin (75–100 mg daily) plus clopidogrel (75 mg daily) is recommended after TAVR only in case of recent coronary stenting (< 3 months), with duration according to bleeding risk (between 1 and 6 months), followed by a lifelong SAPT [2] (Table 1, Figure 6). 

### 5.2. In Antiplatelet Monotherapy, Use of Aspirin or Anti-P2Y12?

Minimal data are available on the optimal SAPT following TAVR. While clopidogrel is also an option in the guidelines, aspirin is commonly used when SAPT is indicated after TAVR. However, whether aspirin or an oral P2Y12 inhibitor should be used remains unknown. Only one observational study compared clopidogrel to aspirin after TAVR. The OCEAN-TAVI Japanese registry reported lower 2-year cardiovascular mortality in patients treated with clopidogrel versus aspirin, with (2.7% vs. 8.5%, respectively) and without (5.2% vs. 18%, respectively) anticoagulation. Neither difference was noticed on all-cause deaths, nor stroke, life-threatening or major bleeding at 2 years [141]. Additional data are needed to evaluate the optimal choice between clopidogrel or aspirin in patients undergoing TAVR with no indication of OAC. However, many studies have highlighted the issue of inhomogeneity in platelet response to clopidogrel in patients undergoing PCI, which has been reported in 20 to 40% of patients. Of note, high platelet reactivity (HPR) has been associated with a higher risk of ischemic events [142]. Limited data are available on the frequency and clinical impact of variable platelet response to antiplatelet therapy in patients undergoing TAVR. While well established in patients experiencing acute coronary syndrome, the safety and efficacy of the oral P2Y12 inhibitor ticagrelor in patients undergoing TAVR needs further investigation. The REAC TAVI (Assessment of Platelet Reactivity after Transcatheter Aortic Valve Implantation) randomized trial compared the efficacy of aspirin plus clopidogrel to aspirin plus ticagrelor after TAVR in patients identified with prior HPR. Among 68 patients, 70% were identify with HPR. Ticagrelor achieved suppression of HPR in all patients, while only 21% of patients treated with clopidogrel had a platelet reactivity reduction [143]. Thus, ticagrelor provides a better and persistent reduction of HPR among patients with aortic stenosis undergoing TAVR. Consistently, the PTOLEMANIOS trial (a trial to assess the safety and efficacy of Prophylactic Ticagrelor With Acetylsalicylic Acid vs. Clopidogrel With Acetylsalicylic Acid in the Development of Cerebrovascular Embolic Events During TAVI) reported better platelet inhibition with ticagrelor (90 mg bd) plus aspirin when compared to clopidogrel plus aspirin, which resulted in fewer suspicious signals of cerebrovascular microembolic events on periprocedural transcranial Doppler [144]. The effect of a lower dose of ticagrelor is currently being investigated in the REACTIC-TAVI (Platelet Reactivity According to Ticagrelor Dose After Transcatheter Aortic Valve Implantation; NCT04331145) trial. This single-arm trial aims to determine the pharmacodynamic effects of 3 months of low-dose ticagrelor (60 mg bd) monotherapy in patients with HPR. Patients with inadequate response to clopidogrel (75 mg qd) are switched to low-dose ticagrelor after TAVR until completing 3 months of treatment, with the efficacy of ticagrelor in suppressing HPR evaluated at different timelines after TAVR.

### 5.3. Patients with Life-Long Indication for Anticoagulation

In patients requiring long-term OAC, previous observational studies have compared outcomes between OAC alone vs. OAC plus antiplatelet therapy (Table 2). Rates of stroke and mortality were similar between the two antithrombotic regimens, while there was an increased risk of bleeding complications with OAC plus APT [48,145,146]. Conversely, the analysis of the PARTNER 2 cohort by Kosmidou et al. questioned the efficacy of OAC alone in the prevention of stroke after TAVR, showing that the 2-year stroke incidence was not reduced with OAC alone, while antiplatelet with or without anticoagulant therapy reduced the risk of stroke at 2 years. However, OAC alone was associated with a reduced risk of combined death and stroke [147]. More recently, cohort B of the POPular-TAVI trial randomized 313 patients to receive OAC (direct oral anticoagulant (DOAC) or VKA) alone or OAC plus clopidogrel for 3 months. The 1-year incidence of bleeding post-TAVI was significantly higher with OAC plus clopidogrel than with OAC alone (34.6% vs. 21.7%, respectively; *p* = 0.01), while the composite endpoint of death from cardiovascular cause and thromboembolic complications (MI and stroke) appeared non-inferior between the two groups [148] (Table 2). Thus, an association of OAC plus antiplatelet therapy leads to a higher rate of bleeding complications with no advantage in long-term thromboembolic complications. Based on these results, OAC alone is recommended by the ESC/EACTS guidelines after TAVR for patients with a lifelong indication of OAC (class of recommendation: I, level of evidence: B), if no concomitant or recent PCI. In the case of coronary stenting in the past 3 months or concomitant to the valve intervention, dual therapy consisting of OAC plus aspirin (or clopidogrel) is recommended for 1 to 6 months according to the bleeding risk, then switched to life-long anticoagulation [2] (Figure 6, Table 1). No specific recommendations are provided in the ACC/AHA Guidelines for patients undergoing TAVR and requiring long-term OAC (Table 1). 

### 5.4. In Patients Requiring OAC, Which Anticoagulant to Choose?

Whether DOAC can be used instead of VKA in patients undergoing TAVR and requiring OAC is a matter of debate. The use of DOACs has been widely approved in patients with nonvalvular AF, with proven noninferiority versus VKA in the prevention of thromboembolic events for dabigatran, rivaroxaban and edoxaban [151,152,153], and superiority for apixaban, with lower rate of bleeding events [154]. In comparison with warfarin, DOACs reduce the risk of stroke, systemic embolism and intracranial hemorrhage in AF patients with valvular heart disease (VHD) (with the exception of severe mitral stenosis or mechanical heart valve) [155]. Furthermore, the North American Consensus Statements have recently been updated and DOAC is the treatment of choice in AF patients undergoing PCI [156]. 

AF is frequent in TAVR patients and associated with poorer outcomes. Furthermore, OAC is a correlated to mortality independently of AF in this population [113]. Observational studies comparing DOACs with VKA provided inconsistent findings [157,158,159,160], and are summarized in Table 3. In the combined France-TAVI and France-2 registries, 8962 patients were treated with OAC after TAVR (24% on DAOCs, 77% on VKAs). After 3 years of follow-up, after propensity matching, there was a significant increase of 37% in mortality rates (VKA vs. DOAC: 35.6% vs. 31.2%; *p* < 0.005) and 64% in major bleeding (12.3% vs. 8.4%; *p* < 0.005) with VKAs compared to DOACs. No between-group difference on ischemic stroke and acute coronary syndrome was reported [161]. Consistently, the STS/ACC registry, the largest to date with 21,131 patients undergoing TAVR with pre-existing or incident AF discharged on OAC, demonstrated a significantly lower incidence of death in patients on DOACs versus VKAs (15.8% vs. 18.2%, respectively; adjusted hazard ratio (HR) 0.92; 95% confidence interval (CI), 0.85–1.00; *p* = 0.043). In addition, DOACs were also associated with a 19% decrease in bleeding rates compared to VKAs (11.9% vs. 15.0%, respectively; adjusted HR 0.81; 95% CI, 0.33–0.87; *p* < 0.001). The 1-year incidence of ischemic stroke was similar between the two groups [162]. Moreover, in a recent meta-analysis of 12 studies, including patients with an indication for OAC, DOACs were associated with lower all-cause mortality compared to VKAs after more than 1 year of follow-up (RR = 0.64; 95% CI 0.42–0.96; *p* = 0.03), while no between-group difference was shown in stroke and valve thrombosis rates [163]. Conversely, in one nonrandomized study, the composite outcome of any cerebrovascular events, myocardial infarction and all-cause mortality was 44% higher in the DOAC group vs. VKA (21.2% vs. 15.0%, respectively; HR 1.44; *p* = 0.05). Nevertheless, the 1-year incidence of all-cause mortality was comparable between the two groups (16.5% vs. 12.2% for DOAC and VKA, respectively; HR: 1.36; 95% CI: 0.90–2.06; *p*= 0.136). Bleeding rates were also similar between the two groups [164]. Of note, the increase of ischemic events was of borderline statistical significance. These findings could be the result of heterogeneity in baseline and procedural characteristics, considering the absence of randomization, and the higher prevalence or renal impairment and peripheral vascular among patients treated with NOAC. Furthermore, previously described observational studies and large registries did not correlate with these findings. Thus, in observational and large registries, DOACs seems to provide similar efficacy to VKA in the prevention of stroke, but improvement in the rates of mortality and bleeding, which could favor the use of DOACs after TAVR in patients with an underlying indication of OAC. However, these data from non-randomized studies should be carefully interpreted.

Recent RCTs did not confirm the superiority of DOACs over VKA on mortality and bleeding outcomes. Only two randomized controlled trials evaluated the use of DOACs after TAVR in patients with an indication of anticoagulant therapy (Table 2). In the ATLANTIS trial (Anti-Thrombotic strategy to Lower All cardiovascular and Neurologic ischemic and hemorrhagic events after Trans-aortic valve Implantation for aortic Stenosis), among the 451 patients with an indication of OAC, results showed no superiority of apixaban compared to VKA on the composite primary outcome of death, thromboembolic events and major bleeding after 1 year of follow-up, with an equivalent incidence of 21.9% in the two groups. No difference was shown between apixaban and VKA on life-threatening or disabling or major bleeding (as defined by VARC-2 consensus document), with an incidence of 10.3% among patients treated with apixaban vs. 11.4% with VKA, (HR 0.92; 95% CI: 0.52–1.60) [115]. Furthermore, the large ENVISAGE-TAVI AF trial (Edoxaban Compared to Standard Care After Heart Valve Replacement Using a Catheter in Patients with Atrial Fibrillation) randomized 1426 post-TAVI patients with AF to either edoxaban (60 mg daily) or VKA, and approximately half of the patients received concomitant antiplatelet therapy (46% in the edoxaban group and 50% in the VKA group). Edoxaban was non-inferior to VKA on the composite primary outcome of all-cause mortality, thromboembolic complications or major bleeding (HR, 1.05; 95% CI 0.85–1.31; *p*= 0.01), but was associated with a higher incidence of major bleeding (40% increase in the edoxaban group), mainly due to gastrointestinal bleeding. In patients treated with concomitant antiplatelet therapy, higher bleeding rates were observed with edoxaban compared to VKA [149]. Thus, edoxaban should be used with caution in TAVR patients requiring long-term OAC, with a rigorous individualized assessment of bleeding risk in each patient, and its association to antiplatelet therapy should be avoided. Nevertheless, data from the ATLANTIS trial support the use of apixaban in patients requiring long-term OAC. Further studies are warranted to clarify the use of DOACs in TAVI patients, and the safety of concomitant antiplatelet therapy if needed.

### 5.5. Patients without Underlying Indication for Anticoagulation

Considering the high risk of thromboembolic complications following TAVR and the potential development of subclinical obstructive valve thrombosis, post-TAVR OAC has been tested in the absence of other indications for anticoagulation. The GALILEO trial compared 3 months administration of low dose rivaroxaban (10 mg daily) plus aspirin followed by rivaroxaban alone with 3 months of aspirin plus clopidogrel followed by aspirin alone. The trial was terminated prematurely, with treatment with rivaroxaban being associated with a significantly higher risk of all-cause death (increase of 69%), of thromboembolic complications and of VARC-2 major, disabling or life-threatening bleeding (increase of 50%), compared to antiplatelet therapy (Table 2) [150]. Consistently, in the Stratum 2 of the ATLANTIS trial, treatment with apixaban resulted in higher all-cause and non-cardiovascular mortality compared with SAPT or DAPT among patients without an indication of OAC (Table 2) [115]. Similar results were reported in An et al. meta-analysis [162]. In the ESC/EACTS guidelines, OAC is contraindicated after TAVR in patients with no indications of OAC (class of recommendation: III, level of evidence: B) (Table 1) [2]. 

### 5.6. Ongoing Trials Evaluating Antithrombotic Therapy after TAVR

The main ongoing trials evaluating the antithrombotic regimen in TAVR patients are summarized in Figure 7. The AVATAR (Anticoagulation Alone Versus Anticoagulation and Aspirin Following Transcatheter Aortic Valve Interventions) open-label randomized controlled trial (*n* = 170) will evaluate the safety and efficacy of anticoagulant therapy alone (VKA or DOAC) versus anticoagulant plus aspirin (NCT02735902) in patients requiring OAC who underwent successful TAVR, after a 12-month follow-up. This trial is expected to end in April 2023. The POPular PAUSE TAVI randomized trial previously cited will provide information on the optimal anticoagulant strategy to adopt during the TAVR procedure, with a comparison of the effect of peri-operative discontinuation versus continuation of OAC on VARC-2 ischemic and bleeding outcomes in patients undergoing TAVR with prior OAC therapy (NCT04437303). Finally, in low-risk patients undergoing TAVR with no indication for OAC, the ongoing LRT (Strategies to Prevent Transcatheter Heart Valve Dysfunction in Low Risk Transcatheter Aortic Valve Replacement) trial is currently exploring the effect of VKA in addition to aspirin, compared to aspirin only, on clinical outcomes and valvular heart deterioration (NCT03557242, with the estimation completion date in July 2023).

## 6. Conclusions

TAVR is expanding towards a low-risk patient category as a result of technical advances and operators’ improved skills. However, the post-TAVR antithrombotic regimen remains challenging. Single antiplatelet therapy appears to be the best compromise when there is no compelling indication for chronic oral anticoagulation. Whether it should be aspirin or clopidogrel is not established. There is no supportive evidence to use oral anticoagulation when there is no established indication for oral anticoagulation other than the TAVR procedure. The gap in evidence as to whether DOACs should be preferred over VKA remains when there is an indication for OAC use. It seems that DOACs are not the same and randomized trials are awaited. Likewise, whether oral anticoagulant therapy should be continued or interrupted during the procedure remains unclear.

## Figures and Tables

**Figure 1 jcm-11-02190-f001:**
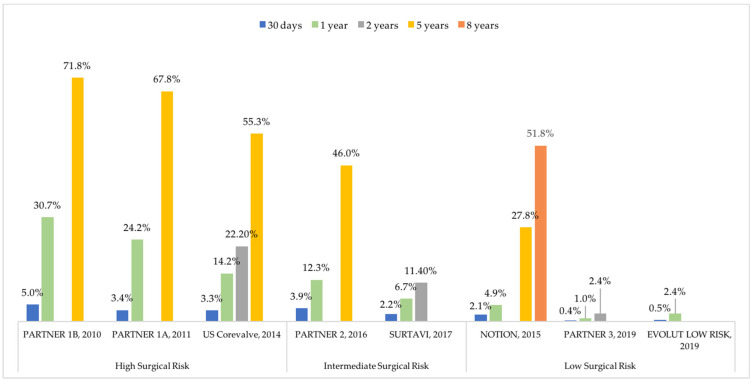
Reported incidences of all-cause mortality after transcatheter aortic valve replacement in pivot trials. NOTION: Nordic Aortic Valve Intervention Trial; PARTNER: Placement of Aortic Transcatheter Valve; SURTAVI: Safety and Efficacy Study of the Medtronic Corevalve© System in the Treatment of Severe, Symptomatic Aortic Stenosis in Intermediate Risk Subjects Who Need Aortic Valve Replacement.

**Figure 2 jcm-11-02190-f002:**
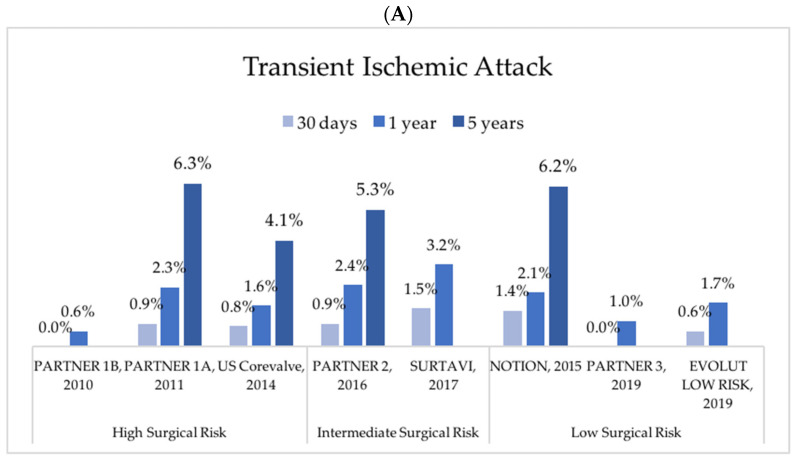
Reported incidences of transient ischemic attack (**A**), major stroke (**B**) and all stroke (**C**) after transcatheter aortic valve replacement in pivot trials. NOTION: Nordic Aortic Valve Intervention Trial; PARTNER: Placement of Aortic Transcatheter Valve; SURTAVI: Safety and Efficacy Study of the Medtronic Corevalve© System in the Treatment of Severe, Symptomatic Aortic Stenosis in Intermediate Risk Subjects Who Need Aortic Valve Replacement.

**Figure 3 jcm-11-02190-f003:**
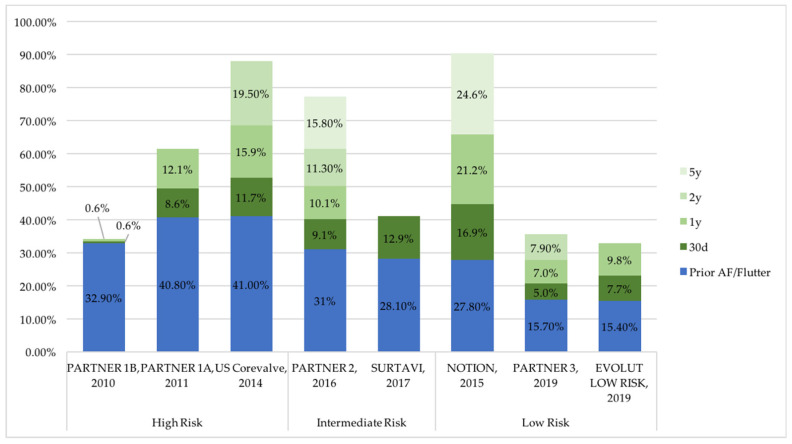
Reported incidences of prior or new onset of atrial fibrillation (AF) after transcatheter aortic valve replacement in pivot trials. NOTION: Nordic Aortic Valve Intervention Trial; PARTNER: Placement of Aortic Transcatheter Valve; SURTAVI: Safety and Efficacy Study of the Medtronic Corevalve© System in the Treatment of Severe, Symptomatic Aortic Stenosis in Intermediate Risk Subjects Who Need Aortic Valve Replacement.

**Figure 4 jcm-11-02190-f004:**
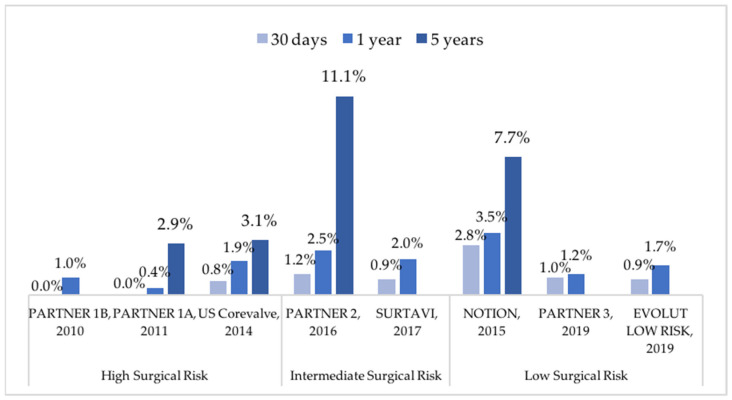
Reported incidences of myocardial infarction after transcatheter aortic valve replacement in pivot trials. NOTION: Nordic Aortic Valve Intervention Trial; PARTNER: Placement of Aortic Transcatheter Valve; SURTAVI: Safety and Efficacy Study of the Medtronic Corevalve© System in the Treatment of Severe, Symptomatic Aortic Stenosis in Intermediate Risk Subjects Who Need Aortic Valve Replacement.

**Figure 5 jcm-11-02190-f005:**
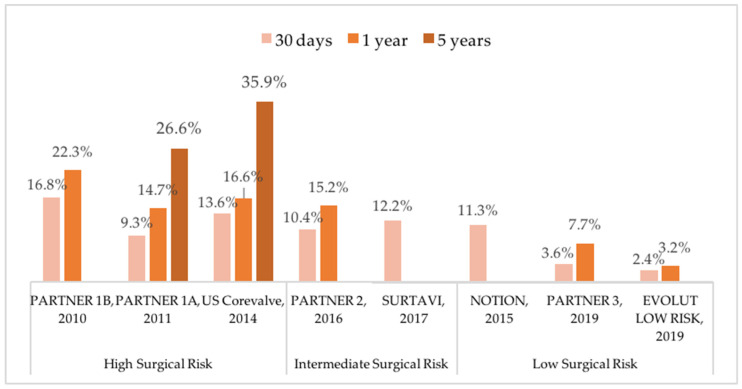
Reported incidences of life-threatening or disabling bleeding after transcatheter aortic valve replacement in pivot trials. NOTION: Nordic Aortic Valve Intervention Trial; PARTNER: Placement of Aortic Transcatheter Valve; SURTAVI: Safety and Efficacy Study of the Medtronic Corevalve© System in the Treatment of Severe, Symptomatic Aortic Stenosis in Intermediate Risk Subjects Who Need Aortic Valve Replacement.

**Figure 6 jcm-11-02190-f006:**
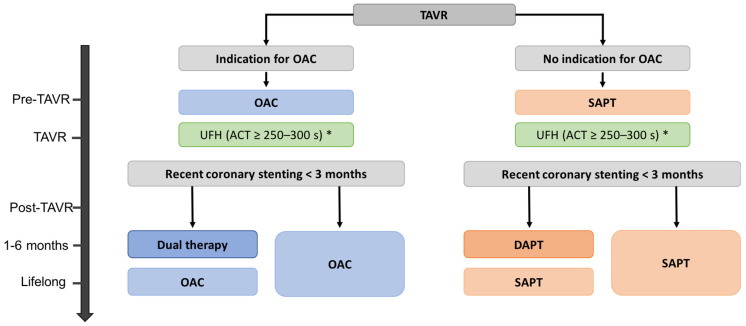
ESC/EACTS recommendations for antithrombotic strategy during and after transcatheter aortic valve replacement. Dual therapy: OAC plus aspirin or clopidogrel. SAPT: Low-dose aspirin or clopidogrel. DAPT: Low-dose aspirin and clopidogrel. OAC: VKA or DOAC. * Bivalirudin if heparin induced thrombocytopenia. ACT: activated clotting time; DAPT: dual antiplatelet therapy; DOAC: non-vitamin K antagonist oral anticoagulant; OAC: oral anticoagulation; SAPT: single antiplatelet therapy; TAVR: transcatheter aortic valve replacement; UFH: unfractioned heparin; VKA: vitamin K antagonist.

**Figure 7 jcm-11-02190-f007:**
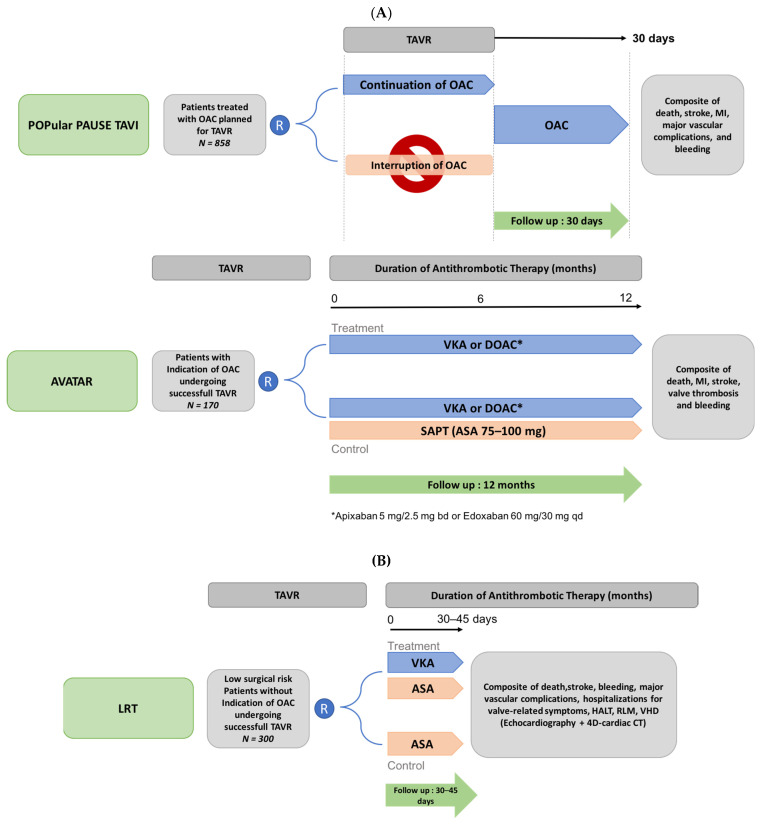
Design of ongoing trials of antithrombotic strategies in patients undergoing transcatheter aortic valve replacement with (**A**) and without (**B**) an indication for oral anticoagulation. 4D-CT: 4-dimensional computed tomography; ASA: aspirin; DOAC: direct oral anticoagulation; HALT: hypoattenuated leaflet thickening; MI: myocardial infarction; RLM: reduced leaflet motion; TAVR: transcatheter aortic valve replacement; VHD: valvular hemodynamic deterioration; VKA: Vitamin K antagonist; bd: bi-day; qd: quotidianly. (**A**) Patients with an underlying indication for OAC. (**B**). Patients without having an indication for OAC.

**Table 1 jcm-11-02190-t001:** Current Recommendations on Antithrombotic Therapy after TAVR.

Guidelines and Expert Consensus	Recommendations	Class of Recommendation	Level of Evidence
ESC/EACTS 2021 Guidelines [2]			
Patients without underlying indication for chronic OAC			
	Lifelong single antiplatelet therapy (aspirin 75–100 mg daily or clopidogrel 75mg daily) is recommended after TAVR in patients with no baseline indication for OAC	I	A
	Routine use OAC is not recommended in patients with no baseline indication for OAC	III	B
Patients with underlying indication for chronic OAC			
	OAC is recommended lifelong for TAVR patients who have other indications for OAC	I	B
**AHA/ACC 2020 Guidelines [3]**			
Patients without underlying indication for chronic OAC			
	For patients with a bioprosthetic TAVR, aspirin 75–100 mg daily is reasonable in the absence of other indications for oral anticoagulants.	IIa	B-R
	For patients with a bioprosthetic TAVR who are at low risk of bleeding, dual antiplatelet therapy with aspirin 75–100 mg and clopidogrel 75 mg may be reasonable for 3–6 months after valve implantation.	IIb	B-NR
	For patients with a bioprosthetic TAVR who are at low risk of bleeding, anticoagulation with a VKA to achieve an INR of 2.5 may be reasonable for at least 3 months after valve implantation.	IIb	B-NR
	For patients with a bioprosthetic TAVR, treatment with low-dose rivaroxaban (10 mg daily) plus aspirin (75-100 mg) is contraindicated in absence of other indications for oral anticoagulants.	III	B-R
Patients with underlying indication for chronic OAC			
	No specific recommendation		
**CCS 2019 Position Statement [88]**			
Patients without underlying indication for chronic OAC			
	Lifelong aspirin 75–100 mg daily	Expert consensus	
	In patients with a recent PCI, dual antiplatelet therapy (aspirin 75–100 mg/d plus clopidogrel 75 mg/d) may be continued as per the treating physician	Expert consensus	
Patients with underlying indication for chronic OAC			
	DOAC for patients with atrial fibrillation unless contra-indicated* in addition to aspirin for TAVR patients	Expert consensus	
	Oral anticoagulation for other indications as per standard guidelines	Expert consensus	
	It is prudent to avoid triple therapy in patients at increased risk of bleeding.	Expert consensus	
**ACCF/AATS/SCAI/STS 2012 Expert Consensus [89]**			
Patients without underlying indication for chronic OAC			
	Antiplatelet therapy for at least 3–6 months after TAVR is recommended to decrease the risk of thrombotic or thromboembolic complications	Expert consensus	
Patients with underlying indication for chronic OAC			
	In patients treated with warfarin, a direct thrombin inhibitor, or factor Xa inhibitor, it is reasonable to continue low-dose aspirin, but other antiplatelet therapy should be avoided, if possible	Expert consensus	
**ACCP-2012 Clinical practice guidelines [90]**			
Patients without underlying indication for chronic OAC			
	Aspirin (50–100 mg/d) plus clopidogrel (75 mg/d) over VKA therapy and over no platelet therapy in the first 3 months	2	C
Patients with underlying indication for chronic OAC			
	No specific recommendation		

* Warfarin would be preferable for patients with contraindication to DOAC in the setting of mitral valve stenosis or mechanical valve replacement. AATS, American Association for Thoracic Surgery; ACC, American College of Cardiology; ACCF, ACC Foundation; ACCP, American College of Chest Physicians; AHA, American Heart Association; CCS, Canadian Cardiovascular Society; DOAC, direct oral anticoagulation; EACTS, European Association for Cardio-Thoracic Surgery; ESC, European Society of Cardiology; INR, international normalized ratio; OAC, oral anticoagulation; PCI, percutaneous coronary intervention; SCAI, Society for Cardiovascular Angiography and Interventions; STS, Society of Thoracic Surgeons; TAVR, transcatheter aortic valve replacement; VKA, vitamin K antagonist.

**Table 2 jcm-11-02190-t002:** RCTs Evaluating Antithrombotic therapy after TAVR.

Trials	First Authors	Compared Strategies	Population	N	Primary End Point	Timeline	Main Results
Patients with an indication of long-term OAC							
**ATLANTIS Stratum 1,** 2021	Collet et al. [115]	Apixaban (5 mg bid) vs. VKA	Successful TAVR with indication for OAC	451	Efficacy: Composite of death, stroke, MI, systemic emboli, intracardiac or valve thrombosis, DVT/PE.	13 mo	Primary end point: Apixaban is not superior to VKA (21.9% vs. 21.9%, respectively; *p* = NS).
					Safety: life-threatening, disabling or major bleeding (VARC-2)		Safety end point: No significant difference between apixaban and VKA (10.3% vs. 11.4%, respectively; *p* = NS)
**ENVISAGE-TAVI AF,** 2021	Van Mieghem et al. [149]	Edoxaban (60 mg qd) vs VKA	Indication for OAC for prevalent or incident AF and sucessful TAVR	1426	Efficacy: Composite of death from any cause, MI, ischemic stroke, systemic thromboembolism, valve thrombosis, or major bleeding.		Primary end point: Edoxaban is not inferior to VKA (17.3 per 100 person-years vs. 16.5 per 100 person-years, respectively (HR 1.05, *p* = 0.01).
					Safety: major bleeding (BARC)		Safety end point: 9.7 per 100 person-years vs. 7.0 per 100 person-years.
**POPULAR-TAVI** **Cohort B,** 2020	Nijenhuis et al. [148]	OAC (VKA or DOACs) vs. OAC + clopidogrel 75 mg 3 mo	Successful TAVR and indication for long term OAC	313	Efficacy: VARC-2 (procedure related) and BARC (non-procedure related) defined bleeding	12 mo	Efficacy: OAC alone is superior to OAC + Clopidogrel (21.7% vs. 34.6%, *p* =0.01).
					Safety: Death from cardiovascular cause and thromboembolic complications		Safety: Similar rates of death from cardiovascular cause and thromboembolic complications (17.3% vs. 13,4% or OAC alone vs. OAC + APT, respectively; *p* =NS)
Patients with no indication of long-term OAC							
**GALILEO,** 2020	Dangas et al. [150]	Rivaroxaban (10 mg qd) + 3-mo ASA (75–100 mg qd) vs. ASA + 3-mo clopidogrel (75 mg qd)	Sucessful TAVR with no indication for OAC	1644	Efficacy: Composite of death or thrombo-embolic events (stroke, MI, symptomatic valve thrombosis, systemic embolism, DVT/PE).	17 mo	Efficacy: 9.8 vs. 7.2 per 100 person-years for rivaroxaban group vs. antiplatelet group, respectively (HR 1.35, *p* = 0.04).
					Safety: VARC-2 defined life-threatening, disabling or major bleeding		Safety: 4.3 vs. 2.8 per 100 person-years, respectively (HR 1.5, *p* = 0.08)
**ATLANTIS Stratum 2,** 2021	Collet et al. [115]	Apixaban (5 mg bid) vs. APT/DAPT	Successful TAVR with no indication for OAC	1049	Efficacy: Composite of death, stroke, MI, systemic emboli, intracardiac or valve thrombosis, DVT/PE.	13 mo	Efficacy: No significant difference between apixaban and VKA (16.9% vs. 19.3%, respectively; *p* = NS).
					Safety: life-threatening, disabling or major bleeding (VARC-2)		Safety: No significant difference between apixaban and VKA (7.8% vs. 7.3%, respectively; *p* = NS)

APT: antiplatelet therapy; ASA: aspirin; DAPT: dual antiplatelet therapy; DVT: deep venous thrombosis; PE: pulmonary embolism; DOAC: non-vitamin K antagonist oral anticoagulant; HR: hazard ratio; MI: myocardial infarction; NS: non-significant; OAC: oral anticoagulation; SAPT: single antiplatelet therapy; TAVR: transcatheter aortic valve replacement; VARC: Valve Academic Research Consortium; VKA: vitamin K antagonist. ATLANTIS: Anti-Thrombotic strategy to Lower All cardiovascular and Neurologic ischemic and hemorrhagic events after Trans-aortic valve Implantation for aortic Stenosis; ENVISAGE-TAVI AF: Edoxaban Compared to Standard Care After Heart Valve Replacement Using a Catheter in Patients with Atrial Fibrillation; GALILEO: Global Study Comparing a Rivaroxaban-based Antithrombotic Strategy to an Antiplatelet-based Strategy After Transcatheter Aortic Valve Replacement to Optimize Clinical Outcomes; POPULAR TAVI: Antiplatelet Therapy for Patients Undergoing Transcatheter Aortic Valve Implantation.

**Table 3 jcm-11-02190-t003:** Main Observational Studies Evaluating Antithrombotic Therapy After TAVR.

First Authors	Year	Compared Strategies	Population	Primary End-Point	Time-Line	Main Results
DOAC vs. VKA						
Tanawuttiwat et al. [162]	2021	DOAC vs. VKA	21,131 patients with indication of OAC, from the STS/ACC TVT Registry	Stroke	1 y	No significant difference on 1-year stroke rates (2.51% vs 2.37% for DOAC and VKA respectively, *p* = 0.980). Lower rate of 1-year bleeding, intracranial hemorrhage and mortality with DOAC compared to VKA.
Didier et al. [161]	2021	DOAC vs. VKA	8962 patients treated with OAC from France-TAVI, France-2 registries and French database	Efficacy: death from any cause. Safety: major bleeding, including hemorrhagic stroke	3 y	DOAC associated with reduced mortality (35.6% vs. 31.2% for DOAC vs VKA, *p* < 0.005) and major bleeding (12.3% vs 8.4%, *p* < 0.005) compared with VKA. No difference on ischemic stroke and acute coronary syndrome.
Butt et al. [159]	2021	DOAC vs. VKA	735 patients with a history of AF	Arterial thromboembolism (composite of ischemic stroke, TIA, thrombosis or embolism in peripheral arteries), all-cause mortality and bleeding	3 y	No significantly different rate of arterial TE (HR 1.23; 95% CI 0.58–2.59), bleeding (HR 1.14; 95%CI 0.63–2.06) or all-cause mortality (HR 0.93; 95%CI 0.61–1.41) after adjustment between DOACs and VKA.
Kawashima et al. [160]	2020	DOAC vs. VKA	403 patients with AF from the OCEAN-TAVI registry	All-cause mortality	Median follow up: 568 d	DOAC was significantly associated with reduced all-cause mortality (10.3% vs 23.3% for DOAC vs. VKA, respectively; *p* = 0.005). No significant difference on VARC-2 defined bleeding and stroke.
Kalogeras et al. [158]	2020	Warfarine vs. DOAC	217 patients with indication of OAC, from the ATLAS registry	All-cause mortality	30 d, 1 y and 2 y	Efficacy: no significant difference on mortality at 30d, 1y and 2y. Safety: no significant difference on the BARC defined major or life-threatening bleeding (10.3% vs. 23.3% for DOAC vs. VKA, respectively; *p* = 0.005).
Mangner et al. [134]	2019	DOAC vs. interrupted VKA (iVKA) or continued VKA (cVKA)	598 patients with AF and on OAC at admission	Early safety VARC-2 criteria	30 d and 1 y	VARC-2 composite criteria lowest with DOAC (13.2%), and not increase in cVKA (19.7%) compared to iVKA (23.1%) (*p* = 0.029). Lowest 1-year mortality with DOAC. No difference on life-threatening of major bleeding.
Jochheim et al. [164]	2019	DOAC or VKA	962 patients with indication of OAC	Composite of all-cause mortality, MI, and any cerebrovascular event	1 y	Higher significant risk with DOAC compared to VKA (21.2% vs. 15%, respectively; HR 1.44, *p* = 0.050). No significant difference on 1-year mortality (16/5% vs. 12.2%, *p* = 0.0136). No significant difference on 1-year bleeding events.
Geis et al. [157]	2018	DOAC vs. VKA	326 patients with underlying indication for OAC	Composite of death, stroke, embolism and severe bleeding (VARC-2)	6 mo	No significant difference (11% vs 8% for DOAC vs. VKA respectively; *p* = 0.45).
Seeger et al. [165]	2017	Apixaban (2.5 mg bid) vs. VKA	272 patients with AF	Early safety VARC-2 criteria at 30-d. Secondary outcome: mortality and stroke at 12 mo	30 day and 12 mo	Early safety end point significantly lower in patients with apixaban compared to VKA (13.5% vs. 30.5% respectively, *p* < 0.01). Reduced risk of life-threatening bleeding with apixaban (3.5% vs. 5.3% respectively, *p* < 0.01). No difference on ischemic outcomes at 12 mo.
**OAC vs. OAC + APT**						
Sherwood et al. [48]	2021	OAC+APT or APT alone or OAC alone	11382 patients with a history of AF	Stroke, all-cause mortality and bleeding events	1 y	After adjustment, no significant difference for all-cause mortality and stroke between the 3 antithrombotic strategies. Similar risk of bleeding when comparing APT alone with OAC alone, and OAC alone with OAC+APT.
Kosmidou et al. [147]	2019	OAC vs. OAC + APT	933 patients with a history of AF and indication for OAC (CHADsVASc ≥ 2) from the PARTNER II trial	Stroke and composite of death and stroke at 2y (VARC-2 definitions)	2 y	After adjustment, significant reduction of stroke and death or stroke with OAC + APT or APT alone compared with no OAC or OAC alone.
Geis et al. [146]	2017	VKA vs. VKA + SAPT/DAPT	167 patients with AF and VKA prescription post TAVR	Composite of death, stroke, thromboembolism and major bleeding	6 mo	Primary end-point less frequent with VKA alone than VKA + SAPT (6.5% vs. 22%; *p* = 0.02) or VKA + DAPT (6.5% vs. 28.6%; *p* = 0.002).
Abdul-Jawad Altisent et al. [145]	2016	VKA vs. VKA + SAPT/DAPT	621 patients with AF with prior VKA therapy	Composite of CV death, MI, stroke and bleeding according to BARC and VARC-2 definitions	13 mo	No difference on ischemic outcomes and death. Major or LTB higher with VKA + SAPT/DAPT compared to VKA alone (24.4% vs. 14.9% respectively, adjusted HR 1.85, *p* = 0.04).

ACC: American College of Cardiology; AF: atrial fibrillation; APT: antiplatelet therapy; ASA: aspirin; 95% CI: 95% confidence interval; CV: cardiovascular; DAPT: dual antiplatelet therapy; DVT: deep venous thrombosis; DOAC: non-vitamin K antagonist oral anticoagulant; HR: hazard ratio; LTB: life-threatening bleeding; MI: myocardial infarction; OAC: oral anticoagulation; SAPT: single antiplatelet therapy; STS: Society of Thoracic Surgeons; TAVR: transcatheter aortic valve replacement; TAVI: transcatheter aortic valve implantation; TE: thromboembolism; TIA: transient ischemic attack; VKA: vitamin K antagonist.

## Data Availability

Not applicable.

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
