# Peer review of "Antithrombotic Therapy Following Transcatheter Aortic Valve Replacement"

_jcm, 2022, doi:10.3390/jcm11082190_

Round 1
Reviewer 1 Report
well done review. Bit too long the first part about strke and myocardial Infarction
Author Response
We would like to thank the reviewer for this comment. We understand the part about ischemic complications seems long, but we aimed to report precisely the latest published data, which is abundant on that matter.
Reviewer 2 Report
It is a timely and comprehensive review.
It is well written and easy to follow.
Here are some suggestions:
Introduction
Page 1, line 36: replace with “...and low surgical risk patients in 2019 following the results of...”
Page 1, line 43: replace with “...the risk of such adverse events.”
Page 2, line 50: replace with “Nonetheless”
Page 6, line 144: replace with “...but only a few randomised studies have evaluated their ability to...”
Page 7, line 212: replace with “...a duration established according to bleeding risk.”
Page 10, line 239: replace with “...on the leaflet of the bioprosthesis,...”
Page 14, line 423-424: replace with “...remains unknown.”
Page 14, line 428: replace with “Additional data...”
Page 15, line 456: replace with “Patients with life-long indication for anticoagulation”
Page 23, line 625: replace with “...established indication for oral anticoagulation...”
Author Response
is a timely and comprehensive review.
It is well written and easy to follow.
response: We would like to thank the reviewer for these kind comments
Here are some suggestions:
Introduction
Page 1, line 36: replace with “...and low surgical risk patients in 2019 following the results of...”
Page 1, line 43: replace with “...the risk of such adverse events.”
Page 2, line 50: replace with “Nonetheless”
Page 6, line 144: replace with “...but only a few randomised studies have evaluated their ability to...”
Page 7, line 212: replace with “...a duration established according to bleeding risk.”
Page 10, line 239: replace with “...on the leaflet of the bioprosthesis,...”
Page 14, line 423-424: replace with “...remains unknown.”
Page 14, line 428: replace with “Additional data...”
Page 15, line 456: replace with “Patients with life-long indication for anticoagulation
Page 23, line 625: replace with “...established indication for oral anticoagulation...”
Response: We updated the manuscript with the pertinent suggestions and corrections.
Reviewer 3 Report
1) General comments
In their original manuscript entitled “Antithrombotic Therapy Following Transcatheter Aortic Valve Replacement”, the authors reviewed that the most updated findings for antithrombic therapy in the TAVR field. This review is very well organized and described the ongoing trials which would help to solve the current unmet issues. This is an important report and I have only a few minor comments.
2) Specific revision comments:
Minor comments
- Page 7, Line 177, 186 and 195: the term “VARC” has to be consistent with.
- Page 11, Line 301: “Beeding” must be an error.
- Page 12, Line 335: “Figure 1” must be an error.
Author Response
In their original manuscript entitled “Antithrombotic Therapy Following Transcatheter Aortic Valve Replacement”, the authors reviewed that the most updated findings for antithrombic therapy in the TAVR field. This review is very well organized and described the ongoing trials which would help to solve the current unmet issues. This is an important report and I have only a few minor comments.
We thank the reviewer for these kind comments.
2) Specific revision comments:
Minor comments
- Page 7, Line 177, 186 and 195: the term “VARC” has to be consistent with.
- Page 11, Line 301: “Beeding” must be an error.
- Page 12, Line 335: “Figure 1” must be an error.
Response: We apologize about the errors and corrected the manuscript as suggested. We understand the incomprehension about the VARC criteria. We clarified the definitions of VARC events, as the most studies studies presented in the manuscript were based on the VARC-2 criteria. We also present recent updates of these definitions were published in a VARC-3 consensus document in 2021. However, these updated VARC 3 criteria now appear at the end of each section to avoid confusion.